# Autoregressive Quantile Flows for Predictive Uncertainty Estimation

**Phillip Si, Allan Bishop, Volodymyr Kuleshov**
Department of Computer Science, Cornell Tech and Cornell University
{ps789, adb262, vk379}@cornell.edu

## Abstract

Numerous applications of machine learning involve representing probability distributions over high-dimensional data. We propose autoregressive quantile flows, a flexible class of normalizing flow models trained using a novel objective based on proper scoring rules. Our objective does not require calculating computationally expensive determinants of Jacobians during training and supports new types of neural architectures, such as neural autoregressive flows from which it is easy to sample.

We leverage these models in quantile flow regression, an approach that parameterizes predictive conditional distributions with flows, resulting in improved probabilistic predictions on tasks such as time series forecasting and object detection. Our novel objective functions and neural flow parameterizations also yield improvements on popular generation and density estimation tasks, and represent a step beyond maximum likelihood learning of flows.

## 1 Introduction

Reasoning about uncertainty via the language of probability is important in many application domains of machine learning, including medicine (Saria, 2018), robotics (Chua et al., 2018; Buckman et al., 2018), and operations research (Van Roy et al., 1997). Especially important is the estimation of predictive uncertainties (e.g., confidence intervals around forecasts) in tasks such as clinical diagnosis (Jiang et al., 2012) or decision support systems (Werling et al., 2015; Kuleshov and Liang, 2015).

Normalizing flows (Rezende and Mohamed, 2016; Papamakarios et al., 2019; Kingma et al., 2016) are a popular framework for defining probabilistic models, and can be used for density estimation (Papamakarios et al., 2017), out-of-distribution detection (Nalisnick et al., 2019), content generation (Kingma and Dhariwal, 2018), and more. Flows feature tractable posterior inference and maximum likelihood estimation; however, maximum likelihood estimation of flows requires carefully designing a family of bijective functions that are simultaneously expressive and whose Jacobian has a tractable determinant. In practice, this limits the kinds of neural networks that can be used to parameterize normalizing flows and makes many types of networks computationally expensive to train.

This paper takes a step towards addressing this limitation of normalizing flows by proposing new objectives that contribute towards alleviating the computational cost of calculating determinants of Jacobians. Specifically, we argue for training flows using an objective that is different from classical maximum likelihood and is instead based on proper scoring rules (Gneiting and Raftery, 2007), a standard tool in the statistics literature for evaluating the quality of probabilistic forecasts. We show that this objective can be used to train normalizing flows and that it simplifies the computation of Jacobians in certain types of flows.

We introduce *autoregressive quantile flows* (AQFs), a framework that combines the above learning objective with a set of architectural choices inspired by classical autoregressive flows. Quantile flows possess characteristics that represent an improvement over existing flow models—including supporting neural architectures that simultaneously provide fast training and sampling—in addition to the usual benefits of flows (exact posterior inference and density estimation). Interestingly, quantile flows can be interpreted as extensions of quantile functions to multiple dimensions.

We use AQFs as the basis for *quantile flow regression* (QFR), an approach to predictive uncertainty estimation in which a probabilistic model directly outputs a normalizing flow as the predictive distribution. The QFR approach enables neural networks to output highly expressive probabilistic predictions that make very little assumptions on the form of the predicted variable and that improve uncertainty estimates in probabilistic and Bayesian models. In the one-dimensional case, our approach yields *quantile function regression* and *cumulative distribution function regression*, two simple, general, and principled approaches for flexible probabilistic forecasting in regression.

In addition, we demonstrate the benefits of AQFs on probabilistic modeling tasks that include density estimation and autoregressive generation. Across our sets of experiments, we observe improved performance, and we demonstrate properties of quantile flows that traditional flow models do not possess (e.g., sampling with flexible neural parameterizations).

**Contributions.**   In summary, this work (1) introduces new objectives for flow models that simplify the computation of determinants of Jacobians, which in turn greatly simplifies the implementation of flow models and extends the class of models that can be used to parameterize flows. We also (2) define autoregressive quantile flows based on this objective, and highlight new architectures supported by this framework. Finally, (3) we deploy AQFs as part of quantile flow regression, and show that this approach improves upon existing methods for predictive uncertainty estimation.

## 2   BACKGROUND

**Notation.**   Our goal is to learn a probabilistic model $p(x) \in \Delta(\mathbb{R}^d)$ in the space $\Delta(\mathbb{R}^d)$ of distributions over a high-dimensional $x \in \mathbb{R}^d$; we use $x_j \in \mathbb{R}$ to denote components of $x$. In some cases, we have access to features $x' \in \mathcal{X}'$ associated with $x$ and we want to train a forecaster $H : \mathcal{X}' \to \Delta(\mathbb{R}^d)$ that outputs a predictive probability over $x$ conditioned on $x'$.

### 2.1   NORMALIZING FLOWS AND AUTOREGRESSIVE GENERATIVE MODELS

A normalizing flow defines a distribution $p(x)$ via an invertible mapping $f_\theta : \mathbb{R}^d \to \mathbb{R}^d$ with parameters $\theta \in \Theta$ that describes a transformation between $x$ and a random variable $z \in \mathbb{R}^d$ sampled from a simple prior $z \sim p(z)$ (Rezende and Mohamed, 2016; Papamakarios et al., 2019). We may compute $p(x)$ via the change of variables formula $p(x) = \left| \frac{\partial f_\theta(z)^{-1}}{\partial z} \right| p(z)$, where $\left| \frac{\partial f_\theta(z)^{-1}}{\partial z} \right|$ denotes the determinant of the inverse Jacobian of $f_\theta$. In order to fit flow-based models using maximum likelihood, we typically choose $f_\theta$ to be in a family for which the Jacobian is tractable.

A common way to define flows with a tractable Jacobian is via autoregressive models of the form
$$x_j = \tau(z_j; h_j) \qquad\qquad h_j = c_j(x_{<j}),$$
where $\tau(z_j; h_j)$ is an invertible transformer, a strictly monotonic function of $z_j$, and $c_j$ is the $j$-th conditioner, which outputs parameters $h_j$ for the transformer. As long as $\tau$ is invertible, such autoregressive models can be used to define flows (Papamakarios et al., 2019).

### 2.2   EVALUATING FORECASTS WITH PROPER SCORING RULES

A common way to represent a probabilistic forecast in the statistics and forecasting literature is via a cumulative distribution function (CDF) $F : \mathbb{R}^d \to [0, 1]$; any probability distribution can be represented this way, including discrete distributions. Since $F$ is monotonically increasing in each coordinate, when $y$ is one dimensional, we may define its inverse $Q : [0, 1] \to \mathbb{R}$ called the quantile function (QF), defined as $Q(\alpha) = \inf\{x' \in \mathbb{R} \mid F(x') \geq \alpha\}$.

In the statistics literature, the quality of forecasts is often evaluated using proper scoring rules (or proper scores; Gneiting and Raftery (2007)). For example, when predictions take the form of CDFs, a popular scoring rule is the continuous ranked probability score (CRPS), defined for two CDFs $F$ and $G$ as $\mathrm{CRPS}(F, G) = \int_y \left( F(x) - G(x) \right)^2 dx$. When we only have samples $x_1, ..., x_m$ from $G$, we can generalize this score as $\frac{1}{m} \sum_{i=1}^m \int_x \left( F(x) - \mathbb{I}(x - x_i) \right)^2 dx$. Alternatively, we can evaluate the $\alpha$-th quantile $Q(\alpha)$ of a QF $Q$ via the check score $L : \mathbb{R} \times \mathbb{R} \to \mathbb{R}_+$ defined as $L_\alpha(x, f) = \alpha(x - f)$ if $x \geq f$ and $(1 - \alpha)(f - x)$ otherwise. The check score also provides a consistent estimator for the conditional quantile of any distribution.

## 3 TAKING STEPS BEYOND MAXIMUM LIKELIHOOD LEARNING OF FLOWS

Maximum likelihood estimation of flows requires carefully designing a family of bijective functions that are simultaneously expressive and whose Jacobian has a tractable determinant. In practice, this makes flows time-consuming to design and computationally expensive to train. In this paper, we argue for training flows using objectives based on proper scoring rules (Gneiting and Raftery, 2007).

### 3.1 LEARNING SIMPLE FLOWS WITH PROPER SCORING RULES

We begin with the one dimensional setting, where a flow $f_\theta : \mathbb{R} \to \mathbb{R}$ is a bijective mapping that can be interpreted as a QF. Alternatively, the reverse flow $f_\theta^{-1}$ can be interpreted as a CDF. We will use $Q_\theta, F_\theta$ to denote $f_\theta$ and $f_\theta^{-1}$, respectively; our goal is to fit these models from data.

In order to fit models of the cumulative distribution and the quantile function, we propose objectives based on proper scoring rules. We propose fitting models $F_\theta$ of the CDF using the CRPS:

$$L^{(1)}(F_\theta, x_i) := \mathrm{CRPS}(F_\theta, x_i) = \int_{-\infty}^{\infty} \left( F_\theta(x) - \mathbb{I}(x_i \le x) \right)^2 dx. \tag{1}$$

When dealing with a model $Q_\theta$ of the QF, we propose an objective based on the expected check score

$$L^{(2)}(Q_\theta, x_i) := \int_0^1 L_\alpha(Q_\theta(\alpha), x_i) d\alpha, \tag{2}$$

where $L_\alpha$ is a check score targeting quantile $\alpha$. We refer to this objective as the *quantile loss*. This objective has been used previously to train value functions in reinforcement learning as well as conditional distributions in autoregressive models (Dabney et al., 2018a;b). In this paper, we describe its application to modeling aleatoric predictive uncertainties.

The parametric form of $Q_\theta$ or $F_\theta$ can be any class of strictly monotonic (hence invertible) functions. Previous works have relied on affine or piecewise linear functions (Wehenkel and Louppe, 2021), sum-of-squares (Jaini et al., 2019), monotonic neural networks (Huang et al., 2018; Cao et al., 2019), and other models. Any of these choices suits our framework; we provide more details below.

**Equivalence Between the CRPS and Quantile Losses**   So far, we have described two methods for fitting a one-dimensional flow model. Their objectives are actually equivalent.

**Proposition 1.** *For a CDF $F : \mathbb{R} \to [0, 1]$ and $x' \in \mathbb{R}$, the CRPS and quantile losses are equivalent:*

$$L^{(1)}(F, x') = a \cdot L^{(2)}(F^{-1}, x') + b \qquad a, b \in \mathbb{R}, a > 0 \tag{3}$$

This fact appears to be part of statistics folk knowledge, and we have only ever seen it stated briefly in some works. We provide a complete proof in the appendix. See (Laio and Tamea, 2007) for another argument.

If the models $F_\theta, Q_\theta$ are analytically invertible (e.g., they are piecewise linear), we are free to choose fitting the CDF or its inverse. Other representations for $F$ will not lead to analytically invertible models, which require choosing a training direction, as we discuss below.

**Practical Implementation.**   The quantile and the CRPS losses both involve a potentially intractable integral. We approximate the integrals using Monte-Carlo; this allows us to obtain gradients using backpropagation. For the quantile loss, we sample $\alpha$ uniformly at random in $[0, 1]$; for the CRPS loss, we choose a reasonable range of $y$ (usually, centered around $y_i$) and sample uniformly in that range. This approach works well in practice and avoids the complexity of alternative methods such as quadrature (Durkan et al., 2019).

### 3.2 LEARNING HIGH-DIMENSIONAL AUTOREGRESSIVE FLOWS

Next, we extend our approach to high-dimensional autoregressive flows by fitting each conditional distribution with the quantile or the CRPS loss. We start with a general autoregressive flow defined as

$$x_j = \tau(z_j; h_j) \qquad\qquad h_j = c_j(x_{<j}), \tag{4}$$

where $\tau(z_j; h_j)$ is an invertible transformer and $c_j$ is the $j$-th conditioner As in the previous section, we use $Q_{h_j}, F_{h_j}$ to denote $\tau(z_j; h_j)$ and $\tau^{-1}(x_j; h_j)$, respectively.

We train autoregressive flows using losses that decompose over each dimension as follows:

$$\underbrace{\frac{1}{n} \sum_{i=1}^{n} \sum_{j=1}^{d} L^{(1)}[x_{ij}, F_{h_j}]}_{\text{CRPS loss; reverse training}} \qquad \underbrace{\frac{1}{n} \sum_{i=1}^{n} \sum_{j=1}^{d} L^{(2)}[x_{ij}, Q_{h_j}]}_{\text{quantile loss; forwards training}}, \qquad (5)$$

where $L^{(1)}$ and $L^{(2)}$ are, respectively, the CRPS and quantile objectives defined in the previous section and applied component-wise. We average the losses on a dataset of $n$ points $\{x_i\}_{i=1}^{n}$.

**Sampling-Based Training of Autoregressive Flows.**  Crucially, Equation 5 defines not only new objectives but also *new training procedures* for flows. Specifically, it defines a *sampling-based* learning process that contrasts with traditional maximum likelihood training.

In order to train with objectives $L^{(1)}, L^{(2)}$, we perform Monte Carlo sampling to approximate an intractable integral. Thus, we sample a *source* variable ($x$ or $z$) from a uniform distribution and transform it into the other variable, which we call the *target*, after which we compute the loss and its gradient. This is in contrast to maximum-likelihood learning, where we take the observed variable $x$, transform it into $z$, and evaluate $p(z)$ as well as the determinant of the Jacobian.

The choice of source and target variable depends on the choice of objective function. Each choice yields a different type of training procedure; we refer to these as forward and reverse training.

**Forward and Reverse Training.**  In forward training, we sample a $z \sim U([0,1])$, pass it through $Q_{h_j}$, obtaining $y_j$. Thus, at training time, we perform *generation* and our loss based on the similarity between samples and the real data $x$. In contrast, we train standard flows by passing an input $y$ into the inverse flow to obtain a latent $z$ (and no sampling is involved). Our process share similarities with the training of GANs, but we don't rely on a discriminator. The forward training process enables us to support neural architectures that maximum likelihood training does not support, as we discuss below.

In reverse training, we sample $x \sim U([x_{\min}, x_{\max}])$ (where the bounds in $U$ are chosen via a heuristic), and we compute the resulting $z$ via $F_\theta$. The CRPS loss defined on $z$ seeks to make the distribution uniform on average, which is consistent with the properties of inverse transform sampling. In this case $\tau(\cdot; h_j)$ can be interpreted as a CDF function conditioned on $x_{<j}$ via $h_j$.

**On Computing Determinants of Jacobians.**  Interestingly, since quantile flows are trained using variants of the CRPS rather than maximum likelihood, they do not rely on the change of variables formula, and do not require computing the derivative $\partial\tau/\partial z$ for each transformer $\tau(\cdot; h_j)$. This fact has a number of important benefits: (1) it greatly simplifies the implementation of flow-based models, and (2) it may also support families of transformers for which computing derivatives $\partial\tau/\partial z$ would have otherwise been inconvenient.

Note however, that our approach does not fully obviate the need to think about the tractability of Jacobians: we still rely on autoregressive models for whom the Jacobian is triangular. We will explore how to further generalize our approach beyond triangular Jacobians in future work.

## 4   Autoregressive Quantile Flows

Next, we introduce *autoregressive quantile flows* (AQFs), a class of models trained with our set of novel learning objectives. Quantile flows improve over classical flow models by, among other things, supporting neural architectures that simultaneously provide fast training and sampling.

An AQF defines a mapping between $z$ and $x$ that has the form $x_j = \tau(z_j; h_j)$ $h_j = c_j(x_{<j})$, as in a classical flow, with $\tau(z_j; h_j)$ being an invertible transformer and $c_j$ being the $j$-th conditioner.

**Architectures for Quantile Flows.**  Crucially, we are free to select a parameterization for either $\tau(z_j; h_j)$ or its inverse $\tau^{-1}(x_j; h_j)$ and the resulting model will be trainable using one of our

objectives $L^{(1)}, L^{(2)}$. This is in sharp contrast to maximum likelihood, in which we need to learn a model of $\tau^{-1}(x_j; h_j)$ in order to recover the latent $z$ from the observed data $x$ and compute $p(z)$.

This latter limitation has the consequence that in classical flows in which the transformer $\tau^{-1}$ is not invertible (which is in most non-affine models), it is difficult or not possible to perform sampling (since sampling requires $\tau$). As a consequence, most non-affine flow models can only be used for density estimation; ours is the first that be used for generation. We give examples below.

**Affine, Piecewise Linear, and Neural Quantile Flows.** Affine transformers have the form $\tau(z; a, b) = a \cdot z + b$, hence they are analytically invertible. A special case is the Gaussian transformer with parameters $\mu, \sigma^2$, which yields the masked autoregressive flow (MAF; Papamakarios et al. (2017)) architecture. The MAF can be trained in our framework using maximum likelihood or $L^{(1)}, L^{(2)}$. Affine flows further extend to piecewise linear transformers, increasing expressivity while maintaining analytical invertiblity. Training affine and piecewise linear transformers provides both a forwards and a reverse mapping between $x$ and $z$ in closed form.

For higher expressivity, we may parameterize transformers with monotonic neural networks, which can be constructed from a composition of perceptrons with strictly positive weights. Huang et al. (2018) proposed an approach where a conditioner $c(x_{<j})$ directly outputs the weights of a monotonic neural network which models $\tau^{-1}(x_j, c(x_{<j}))$. In our experiments, $\tau(z_j, c(x_{<j}))$ is simply parameterized by a network with positive weights that outputs $x_j$ as based on $z_j, x_{<j}$, similarly to (Cao et al., 2019), although AQFs also support the approach of Huang et al. (2018).

**Comparison to Other Flow Families.** In general, any transformer family that can be used within autoregressive models can be used within the quantile flows framework, including integration-based transformers (Wehenkel and Louppe, 2021) spline approximations (Müller et al., 2019; Durkan et al., 2019; Dolatabadi et al., 2020), piece-wise separable models, and others.

A key feature of our framework is that it *supports both forwards and reverse training* while alternatives do not. For example, when training a neural autoregressive flow (Huang et al., 2018), we need to compute $z$ from $x$ to compute the log-likelihood. Thus, we parameterize $\tau^{-1}$ with a neural network, but then $\tau$ is not analytically invertible, which precludes us from sampling from it. Our method, on the other hand, parameterizes $\tau$ itself using a flexible neural approximator, making it possible to easily perform sampling within a flexible neural autoregressive flow for the first time.

Additionally, modeling flexible CDFs enables us to study broader classes of distributions, including discrete ones. We perform early experiments and leave the full extension to future work.

## 5   Predictive Uncertainty Estimation Using Quantile Flows

Probabilistic models typically assume a fixed parametric form for their outputs (e.g., a Gaussian), which limits their expressivity and accuracy (Kuleshov et al., 2018). Here, we use AQFs as the basis for *quantile flow regression* (QFR), which models highly expressive aleatoric uncertainties over high-dimensional variables by directly outputting a normalizing flow.

### 5.1   Quantile Flow Regression

The idea of Quantile Flow Regression (QFR) is to train models $H : \mathcal{X} \to (\mathbb{R}^d \to [0, 1])$ that directly output a normalizing flow. This approach does not impose any further parametric assumptions on the form of the output distribution (besides that it is parameterized by an expressive function approximator), enabling the use of flexible probability distributions in regression, and avoiding restrictive assumptions of the kind that would be made by Gaussian regression.

From an implementation perspective, $H$ outputs quantile flows of the form

$$y_j = \tau(z_j; h_j) \qquad\qquad h_j = c_j(x_{<j}, g(x)), \qquad\qquad (6)$$

where each conditioner $c_i(x_{<i}, g(x'))$ in a function of an input $x'$ processed by a model $g(x')$. We provide specific examples of this parameterization in our experiments section. The entire structure can be represented in a single computational graph and is trained end-to-end using objectives $L^{(1)}, L^{(2)}$.

**Estimating Probabilities from Quantile Flows.** In practice, we may also be interested to estimate the probability of various events under the joint distribution of $x$. This may be done by sampling from the predictive distribution and computing the empirical occurrence of the events of interest. See our time series experiments for details, as well as the appendix for pseudocode of the sampling algorithm.

Note that while we cannot directly estimate events such as $P(x_1 \leq x \leq x_2)$ from a multi-variate quantile flow, we can always compute densities of the form $P(x)$. When fitting flows in the reverse direction, we can simply differentiate each transformer $d\tau/dx$ to obtain conditional densities. When fitting flows in the forward direction, we can use the formula $d\tau/dz = 1/P(\tau(z))$.

## 5.2 Quantile and Cumulative Distribution Function Regression

In the one-dimensional case, our approach yields *quantile function regression* and *cumulative distribution function regression* (CDFR), two simple, general, and principled approaches for flexible probabilistic forecasting in regression. In quantile function regression, a baseline model $H$ outputs a quantile function $Q(\alpha)$; in practice, we implement this model via a neural network $f(x', \alpha)$ that takes as input an extra $\alpha \in [0, 1]$ and outputs the $\alpha$-th quantile of the conditional distribution of $x$ given $x'$. We train the model $f$ with the quantile loss $L^{(2)}$ while sampling $\alpha \sim U([0, 1])$ during the optimization process. We provide several example parameterizations of $f$ in the experiments section. Cumulative distribution function regression is defined similarly to the above and is trained with the CRPS loss $L^{(1)}$.

**Quantile vs. Cumulative Distribution Functions.** The quantile function $Q$ makes it easy to derive $\alpha$-% confidence intervals around the median by querying $Q(1 - \alpha/2)$ and $Q(\alpha/2)$. Conversely, the CDF function $f$ makes it easy to query the probability that $x$ falls in a region $[x_1, x_2]$ via $f(x_2) - f(x_1)$. Each operation can also be performed in the alternative model, but would require bisection or grid search. We suspect users will fit one model or both, depending on their needs.

**Extensions to Classification.** A CDF can also be used to represent discrete random variables; thus our method is applicable to classification, even though we focus on regression. We may fit a CDF $F$ over $K$ numerical class labels $x_1 < ... < x_K$; the optimal form of $F$ is a step function. When $F$ is not a step function, we can still extract from it approximate class membership probabilities. For example, in binary classification where classes are encoded as $0$ and $1$, we may use $(F(1 - \epsilon) - F(0))/2$ as an estimate of the probability of $x = 0$. We provide several experiments in this regime.

## 6 Experiments

We evaluate quantile flow regression and its extensions against several baselines, including Mixture Density Networks (MDN; Bishop (1994)), Gaussian regression, Quantile regression (QRO; in which we fit a separate estimator for 99 quantiles indexed by $\alpha = 0.01, ..., 0.99$). We report calibration errors (as in Kuleshov et al. (2018)), check scores (CHK), the CRPS (as in Gneiting and Raftery (2007)) and L1 loss (MAE); both calibration error (defined in appendix) and check score are indexed by $\alpha = 0.01, ..., 0.99$. Additional experiments on synthetic datasets can be found in the appendix.

### 6.1 Experiments on UCI Datasets

**Datasets** We used four benchmark UCI regression datasets (Dua and Graff, 2017) varying in size from 308-1030 instances and 6-10 continuous features. We used two benchmark UCI binary classification datasets with 512 and 722 instances with 22 and 9 features respectively. We randomly hold out 25% of data for testing.

**Models.** QFR and CDFR were trained with learning rates and dropout rates of $3e-3, 3e-4$ and $(0.2, 0.1)$ respectively as described in Section 5.2. All of the models were two-hidden-layer neural networks with hidden layer size 64, and ReLU activation functions.

**Results.** QFR, on average, outperforms the baselines in each of the one-dimensional UCI tasks (Table 1). Interestingly, our method was also able to learn a CDF for binary classification. We

Table 1: Results on the regression benchmark datasets.

| Dataset | Check Score | | | | | CRPS | | | | |
|---|---|---|---|---|---|---|---|---|---|---|
| | Gaussian | QRO | MDN | QFR | CDFR | Gaussian | QRO | MDN | QFR | CDFR |
| Yacht | 0.225 | 0.202 | 0.174 | **0.171** | - | 0.443 | 0.381 | 0.282 | **0.275** | 0.415 |
| Boston | 1.547 | 1.191 | 1.115 | **0.865** | - | 2.643 | 1.441 | 1.160 | **0.025** | 2.373 |
| Concrete | 2.243 | 1.563 | 2.896 | **1.191** | - | 4.157 | 1.906 | 5.703 | **0.697** | 2.97 |
| Energy | 2.254 | 1.463 | 2.926 | **1.183** | - | 4.171 | 1.911 | 5.745 | **0.578** | 2.91 |

| Dataset | MAE | | | | | Calibration MAE | | | | |
|---|---|---|---|---|---|---|---|---|---|---|
| | Gaussian | QRO | MDN | QFR | CDFR | Gaussian | QRO | MDN | QFR | CDFR |
| Yacht | 0.627 | 0.563 | 0.494 | **0.481** | 0.539 | 0.097 | 0.100 | **0.019** | 0.028 | 0.116 |
| Boston | 3.909 | 3.398 | 2.933 | **2.215** | 2.611 | 0.139 | 0.140 | 0.036 | **0.034** | 0.038 |
| Concrete | 6.329 | 4.051 | 9.022 | **2.853** | 3.623 | 0.165 | 0.114 | **0.029** | 0.046 | 0.066 |
| Energy | 6.629 | 4.151 | 9.393 | **2.815** | 2.829 | 0.165 | 0.114 | 0.032 | **0.029** | 0.049 |

demonstrated that the CDFR method is able to compete with the neural network baseline and that our method can be potentially extended to classification, testing on the Diabetes and KC2 dataset where the Neural Network obtained metrics of 0.717 and 0.832 respectively, and our CDFR obtained 0.723 and 0.796 respectively.

## 6.2 OBJECT DETECTION

**Data.** We also conducted bounding box retrieval experiments against Gaussian baselines on the data corpus VOC 2007, obtained from Everingham et al. with 20 classes of objects in 9963 images.

Table 2: Obj. Detection with CDFR

| Method | CRPS |
|---|---|
| Gaussian | 6.85 |
| CDFR | 6.15 |

**Models.** We modified the architecture of He et al. (2019) which uses a KL-loss to fit a Gaussian distribution for each bounding box. The existing architecture has a MobileNet-v2 backbone (Sandler et al., 2019). Keeping the backbone unchanged, the quantile regressor method adds an alternate head with separate layers, which adds uncertainty bounds to the original output. The original Gaussian method was trained for a total of 60 epochs, with a learning rate of 2e-4 which is decremented by a factor of 10 at epochs 30 and 45. The quantile flow layer is trained for an additional 20 epochs, with the same learning rate.

**Results.** We implement QFR in conjunction in a standard model, as well as one that uses variational dropout (VD; Gal and Ghahramani (2016)). In each case, the QFR layer improves uncertainty estimation (as measured by CRPS; Table 3), with the best results coming from a model trained with QFR and VD. MAP@50 is defined in the appendix. This suggests our methods improve both aleatoric

Table 3: Results on the object detection task

| Method | Cal. MAE | MAP@50 | CRPS |
|---|---|---|---|
| Gaussian | 0.133 | 0.796 | 8.89 |
| QFR | 0.103 | 0.796 | 8.59 |
| Gauss. VD | - | 0.771 | 8.73 |
| QFR VD | - | 0.772 | 7.99 |

and epistemic uncertainties (the latter being estimated by VD). We illustrate examples of predicted uncertainties on real images in Figure 1. We also swapped QFR for CDFR, while keeping approximately the same architecture, and compared to the Gaussian baseline using the CRPS. CDFR also outperforms the Gaussian baseline (Table 2).

## 6.3 TIME SERIES FORECASTING

**Data.** Our time series experiment uses quantile flows to forecast hourly electricity usage. We follow the experimental setup of Mashlakov et al. (2021) using their publicly available code. We use the 2011-2014 Electricity Load dataset, a popular dataset for benchmarking time series models. It contains 70,299 time series derived from hourly usage by 321 clients across three years. There exist different ways to pre-process this dataset; we follow the setup of Mashlakov et al. (2021). Our models predict hourly electricity usage 24 hours into the future for each time series, for a total of 1,687,176 predictions. There are four covariates.

**Models.** We benchmark two state-of-the-art time series forecasting models: DeepAR (Salinas et al., 2019) and MQ-RNN (Wen et al., 2018). Both use an autoregressive LSTM architecture as in

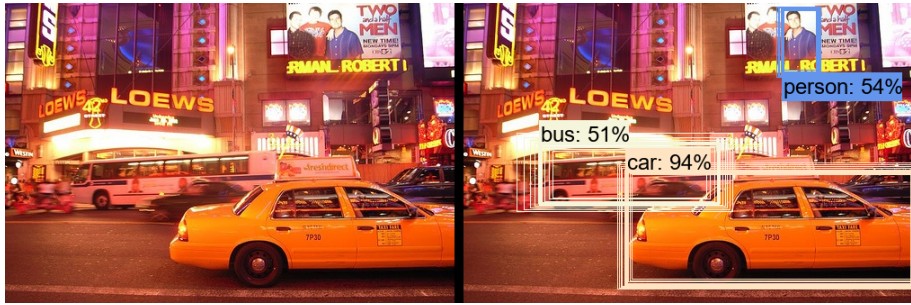

Figure 1: Examples of bounding box uncertainties on the object detection task. The thicker boundary represents the original model prediction, and the thinner lines represent ten quantiles (between 0.1 and 0.9) predicted via QFR. High-confidence boundaries (such as those of the car) are closer together, while low-confidence boundaries (e.g., the bus which is blurred and partially occluded) are asymmetrically spread out, reflecting the advantages of non-Gaussian uncertainty estimation.

Mashlakov et al. (2021); DeepAR outputs Gaussians at each time step, while MQ-RNN estimates five conditional quantiles, which are followed by linear interpolation.

We train a recurrent model with same LSTM architecture and replace the predictive component with a quantile flow. Each transformer $\tau(z_i; h_i)$ is a monotone neural network with one hidden layer of 20 neurons, conditioned on the output $h_i$ of the LSTM module, acting as the conditioner $c_i(y_{<i}, x)$.

**Results.** Figures 2a and 2b compare predictions from DeepAR and AQF (in blue) on illustrative time series (red). We take samples from both models and visualize the $(0.1, 0.9)$ quantiles in shaded blue. While both time series underpredict, the Gaussian is overconfident, while the samples from AQF contain the true (red) curve. Moreover, AQF quantiles expand in uncertain regions and contract with more certainty. We also compare the three methods quantitatively in Table 4. Both achieve similar levels of

Table 4: Time series experiments

| Method | ND | RMSE | CHK |
|---|---|---|---|
| DeepAR | 9.9% | 0.692 | 89.15 |
| MQ-RNN | 9.9% | 0.712 | 88.91 |
| AQF (ours) | 9.8% | 0.723 | 88.02 |

accuracy (ND and RMSE are as defined in Salinas et al. (2019), but we have included them in the appendix as well), and AQF produces better uncertainties. We also found that for 37.2% of time series, DeepAR and AQF performed comparably for CRPS (CRPS within 5% of each other); on 30.3% of time series DeepAR fared better, and on 32.1% AQF did best.

## 6.4 GENERATIVE MODELING

A key feature of AQFs is that they can be used for generative modeling with expressive neural parameterizations since they can be trained in the forward direction, unlike Huang et al. (2018). We benchmark three different methods for generative flows, MAF trained via maximum likelihood (MAF-LL) and with our quantile loss (MAF-QL), and our proposed method, Neural Quantile Flow (NAQF) with the same number of parameters on various UCI datasets. The models consist of an LSTM layer followed by a FC layer, with 40 hidden states in each. The MAFs generate the parameters for a Gaussian distribution from which the results are sampled.

**Results.** Figure 3 shows the error curves of the MAF models trained with log-likelihood and with our loss. Interestingly, there is little correlation between the two, which is reminiscent of other model classes trained with different objectives (e.g., LL-Flows and

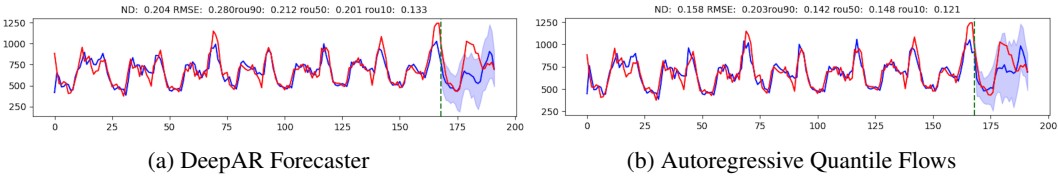

(a) DeepAR Forecaster          (b) Autoregressive Quantile Flows

Figure 2: Examples of Predictive Uncertainties on the Times Series Forecasting Task

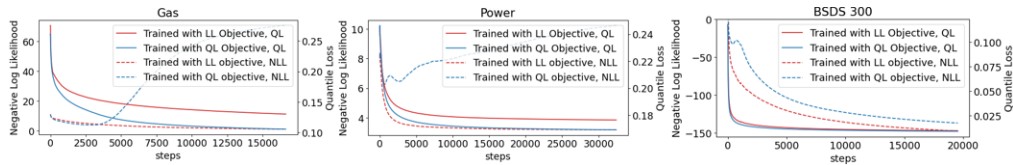

Figure 3: Log likelihood and CRPS error curves for the MAF on three UCI Datasets.

GANs); in some cases, likelihood can even diverge as we train with our loss. Table 5 evaluates CRPS trained with each model. The best results are achieved with the NAQF, even though it uses the same number of parameters, indicating the value of neural transformers.

### 6.5 IMAGE GENERATION

We demonstrate the ability of our model to generate samples using a neural parameterization by generating images of digits from sklearn (Figure 4). We use 5 LSTM layers with 80 neurons; all models are trained to convergence with learning rate 1e-3, with 500 epochs for the NAQF and MAF-QL, while the MAF-LL was given an extra 500 epochs to reach convergence. Additionally, we created a SVM discriminator to differentiate the real data versus the sampled data from the model (D-Loss in Table 6, defined formally in appendix). The better the sampled data is, the lower the accuracy achieved by the discriminator.

Table 5: CRPS on UCI datasets.

| Dataset | MAF-LL | MAF-QL | NAQF |
|---|---|---|---|
| BSDS 300 | .044 | .036 | **.033** |
| Miniboone | .567 | .561 | **.525** |
| Gas | .645 | .565 | **.513** |
| Power | .542 | .506 | **.502** |
| Hepmass | .617 | .614 | **.523** |

**Results.** The NAQF generates significantly better samples that are the most challenging to discriminate from real samples, and also echieves the best CRPS values. The MAF method trained with LL achieves worst results and visibly worse samples, despite having the same number of parameters and being trained for longer. Note that despite the MAF-QL having a considerably worse NLL, the samples generated are still more representative of the data, according to the discriminative model, and visually the sample quality is much higher.

Table 6: Image Generation Experiments

| Method | CRPS | LL | Q-Loss | D-Loss |
|---|---|---|---|---|
| MAF-LL | 2.31 | **-45** | 0.625 | 0.978 |
| MAF-QL | 2.16 | -6187 | 0.281 | 0.779 |
| NAQF | **2.14** | n/a | **0.253** | **0.723** |

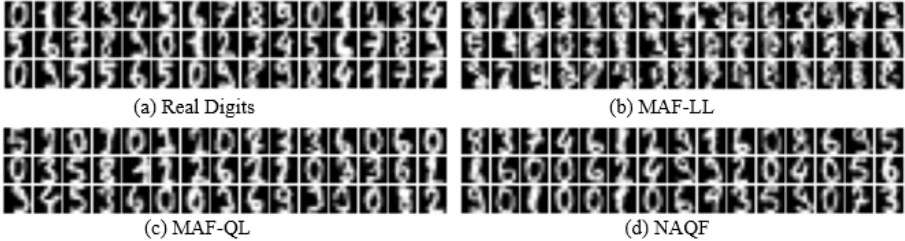

(a) Real Digits   (b) MAF-LL

(c) MAF-QL   (d) NAQF

Figure 4: Images of digits sampled from four autoregressive flow methods.

### 7 DISCUSSION AND CONCLUSION

Our work extends the framework of flows Papamakarios et al. (2017; 2019) to use a novel learning objectives. Similarly, it generalizes implicit quantile networks Dabney et al. (2018b) into a normalizing flow framework. Our work is an alternative to variational (Kingma and Welling, 2014), adversarial (Goodfellow et al., 2014), or hybrid (Larsen et al., 2016; Kuleshov and Ermon, 2017) generative modeling and extend structured prediciton models (Lafferty et al., 2001; Ren et al., 2018). Interestingly, quantile flows can be seen as a generalization of quantile functions to higher dimensions. While several authors previously proposed multi-variate generalizations of quantiles (Serfling, 2002; Liu and Wu, 2009), neither of them possess all the properties of a univariate quantile function, and ours is an additional proposal.

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

## A  UCI DATASETS EXPANDED

**Yacht**. The yacht dataset consists of 308 instances, 6 features and 1 continuous target.
**Boston**. The boston house prices dataset consists of 506 instances, 14 features and 1 continuous target.
**Concrete**. The concrete dataset consists of 1030 instances, 8 features and 1 continuous target.
**Energy**. The energy efficiency dataset consists of 768 instances, 10 features and 1 continuous target.
*Classification Datasets*
**KC2**. The kc2 dataset consists of 522 instances, 22 features and 1 binary target.
**Diabetes**. The diabetes dataset consists of 769 instances, 9 features and 1 binary target.
*Multivariate Datasets*
**ENB**. The enb dataset consists of 768 instances, 10 features and 2 continous targets.

## B  PROOF OF PROPOSITION

**Proposition 2.** *For a CDF $F : \mathbb{R} \to [0, 1]$ and $y \in \mathbb{R}$, the CRPS and quantile losses are equivalent:*

$$L^{(1)}(F, y') = a \cdot L^{(2)}(F^{-1}, y') + b \qquad a, b \in \mathbb{R}, a > 0 \tag{7}$$

*Proof.* Recall that the pinball loss is defined as

$$L_\alpha(y, f) = \alpha(y - f) \text{ if } y \geq f \text{ and } (1 - \alpha)(f - y) \text{ otherwise.}$$

Observe that we can equivalently write

$$2L_\alpha(F^{-1}(\alpha), y') = |F^{-1}(\alpha) - y'| + 2(\alpha - 0.5)(F^{-1}(\alpha) - y')$$

We take the expected pinball loss and perform the change of variables $y = F^{-1}(\alpha)$:

$$2 \cdot \int_0^1 L_\alpha(F^{-1}(\alpha), y')d\alpha = \int_0^1 \left[|F^{-1}(\alpha) - y'| + 2(\alpha - 0.5)(F^{-1}(\alpha) - y')\right] d\alpha$$

$$= \int_0^1 \left[|y - y'| + (2F(y) - 1)(y - y')\right] f(y)dy,$$

where $f(y)$ is the probability density function associated with $F$.

Using integration by parts and the fact that $F(x') - \mathbb{I}(x' \leq x) \to 0$ as $x \to \infty$ and as $x \to -\infty$, we obtain

$$\int_0^1 \left[|x - x'| + (1 - 2F(x))(x - x')\right] f(x)dy = \int_{-\infty}^\infty \left(F(x') - \mathbb{I}(x' \leq x)\right)^2 dx.$$

$\square$

## C  EXTRA FIGURES

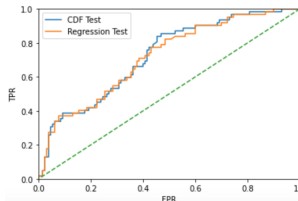

(a) OpenML KC2 ROC           (b) OpenML Diabetes ROC

## D  SYNTHETIC EXPERIMENTS

In addition to the real-world dataset experiments, we also conducted experiments with synthetic datasets, which provides additional insights into the properties of autoregressive quantile flows.

Table 8: Calibration Metrics and Accuracy

| Metric | AQF | Gaussian. |
|---|---|---|
| RMSE | 1.017 | 1.016 |
| Calibration 0.25 | 0.236 | 0.342 |
| Calibration 0.5 | 0.493 | 0.510 |
| Calibration 0.75 | 0.746 | 0.684 |

Table 9: Results on the 1D synthetic data benchmarks.

| Dataset | Check Score | | | | | CRPS | | | | |
|---|---|---|---|---|---|---|---|---|---|---|
| | AQF | Gaussian | MDN | Poisson | True | AQF | Gaussian | MDN | Poisson | True |
| Gaussian | **0.302** | 0.330 | 0.308 | - | 0.294 | **0.586** | 0.600 | 0.610 | - | 0.581 |
| Beta | **0.233** | 0.251 | 240 | - | 0.224 | **0.460** | 0.547 | 0.493 | - | 0.443 |
| Poisson | **0.487** | 0.517 | 0.537 | 0.502 | 0.452 | **0.965** | 1.027 | 1.059 | 0.993 | 0.898 |

### D.1    ONE-DIMENSIONAL EXPERIMENTS

We start with a series of one-dimensional experiments involving the a simple 1D function to which we add additive Gaussian, Beta, and Poisson noise.

The Gaussian and Beta distribution experiments were generated from a base function of $f(x') = \sin(x'/2) + x'/10$ such that $x = f(x') + \mathcal{N}(0,1)$ and $x = f(x') + \mathcal{B}(2,5)$ respectively. The Poisson distribution was generated from $\mathcal{P}(f(x') + C)$, where C is a constant term to make all terms non-negative.

We fit these functions using autoregressive quantile flows (AQFs) as well as the following Baselines: Gaussian regression, Mixture Density Networks (MDNs), and Poisson regression. All models implement the same neural networks and differ only in how the final layer represents aleatoric uncertainty.

We calculated check score and CRPS metrics for each model. In addition, we calculated the metrics for the true distribution, as a upper bound on the performance of the models. As shown in Table 9, AQF is much closer to the true distribution compared to the other models with Gaussian assumptions, especially when performed on non-Gaussian distributions.

This shows that fitting an expressive AQF model *better captures predictive uncertainty* than alternative models that make parametric assumptions on the output distribution. In other words, using a parametric distribution (e.g., Poisson or Gaussian) works well if the underlying data actually follows that distribution. When using an AQF, *we don't need to select a predictive distribution type*, it just works out of the box as if we knew it.

We also report calibration error, defined as the absolute difference between the percentage of points below the predicted quantile curves and the actual quantile value (Table 7). For a perfectly Gaussian noise distribution, the QRF (analogous to AQF in 1-dimension) and QRO methods (QRO consists of fitting a separate estimator for 99 quantiles indexed by $\alpha = 0.01, ..., 0.99$) are comparable to a Gaussian regressor in terms of

Table 7: Calibration Error on 1D datasets

| Dataset | QRF | QRO | Gaussian |
|---|---|---|---|
| Gaussian | 1.01 | 1.03 | **0.952** |
| Beta | **1.39** | 1.75 | 4.63 |
| Poisson | 2.76 | **2.57** | 17.64 |

calibration. When the distributions become non-Gaussian, QRF is adaptable whereas the Gaussian fails to capture the distribution. Thus, the Gaussian maximum likelihood loss tends to be overconfident, a fact also demonstrated in Kuleshov et al. (2018); AQF addresses the overconfidence.

**Supervised Accuracy and Calibration at Key Quantiles**    We also report calibration values at key quantiles (25%, 50%, 75%) in Table 8. We also measure predictive accuracy in terms of RMSE and demonstrate that our method obtains good estimates of uncertainty without sacrificing supervised performance in terms of regression quality.

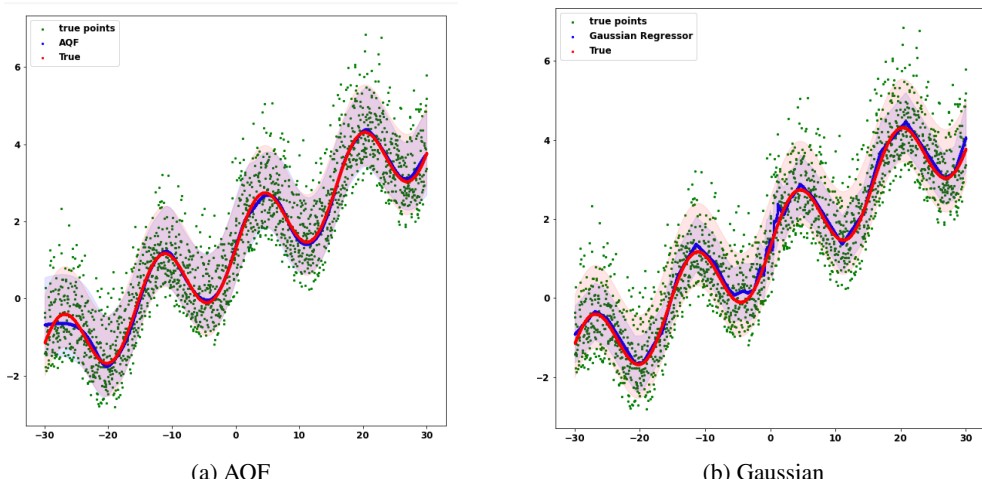

|       (a) AQF       |       (b) Gaussian       |

Figure 6: 80% Confidence Intervals obtained via AQF (left) and Gaussian regression (right). Note that confidence intervals obtained from Gaussian regression (right, blue) differ significantly from the true confidence intervals (right, red). Confidence intervals obtained from AQF (left, blue) perfectly overlap with the true intervals (left, red).

In Figure 6a and 6b, we report 80% confidence intervals for AQF and Gaussian regression methods on the synthetic beta noise data for a visualization of the quantile predictions.

## D.2 MULTI-DIMENSIONAL EXPERIMENTS

We also conducted multi-dimensional experiments with a $d$-dimensional target $x$; we model the first dimension by $x_0 = 5\sin(x'/3) + x' + \mathcal{N}(0,1)$; then, further dimensions are modeled by $y_d = 5\sin(x_{d-1}/3) + x_{d-1} + \mathcal{N}(0,1)$. We use both the marginal CRPS and the joint CRPS metrics, as described in Jordan et al. (2018), for both a simple 2d-experiment as well as a more complex 50-dimensional experiment.

The methods we evaluate are our proposed AQF, a Gaussian Markov Random Field (a type of structured prediction algorithm that assumes a multi-variate Gaussian distribution on the output), and a model in which we train a multi-variate Gaussian distribution using quantile loss instead of log likelihood (denoted as "Quantile" in the table). As in the 1D synthetic experiments, we also compare metrics against samples drawn from the true distribution to obtain an upper bound.

Table 10 shows that AQF does a better job at capturing the high-dimensional distribution than models that assume a more restricted parametric form, such as the Gaussian MRF. Simply replacing the objective function with quantile loss instead of standard Gaussian noise does not provide as significant of a gain as AQF does. In other words, existing models (e.g., a Gaussian MRF) are not sufficiently expressive to capture the multi-variate distribution that he we have defined. In contrast the AQF layer is able to achieve close to approach the theoretical upper bound without making any additional modeling assumptions.

Table 10: Results on Multidimensional Synthetic Datasets

| Dataset | CRPS (Marginal) | | | | CRPS (Joint) | | | |
|---------|-------|----------|----------|-------|-------|----------|----------|-------|
|         | AQF   | Gaussian | Quantile | True  | AQF   | Gaussian | Quantile | True  |
| 2D      | **0.624** | 0.669 | 0.634 | 0.585 | **0.686** | 0.701 | 0.695 | 0.646 |
| 50D     | **1.045** | 1.133 | 1.058 | 0.970 | **1.238** | 1.345 | 1.281 | 1.163 |

# E  ALGORITHM

---

**Algorithm 1** Training a Neural Network $f$ with the Quantile Loss

---

Sample $\alpha = \mathcal{U}(0, 1)$
Sample a training instance $(x', x)$ from the dataset (or a batch of instances)
Compute prediction $y_{\text{pred}} = f(x', \alpha)$
Compute quantile loss $L_\alpha(x, x_{\text{pred}})$.
Take a gradient step quantile loss $L_\alpha(x, x_{\text{pred}})$.

---

# F  ADDITIONAL DEFINITIONS FOR USED METRICS

## F.1  CALIBRATION MAE

If we have a 1-dimensional model which predicts a curve $F(q)$ for quantile q, then we have that our Calibration MAE on test set X is given by

$$\frac{1}{99} \sum_{q=1}^{99} |P(X \le F(q)) - q|$$

## F.2  MAP@50

MAP@50 denotes the mean average precision for an IoU value of 0.5. IoU is computed as, given ground truth boxes G and predicted box B, $\frac{G \cap B}{G \cup B} \ge 0.5$. MAP is then calculated as the AUC of the precision-recall curve given the IoU.

## F.3  RMSE AND ND (TIME SERIES)

We will define data point $z_{i,t}$ to be the t-th time in the i-th context window, and $\hat{z}$ to be the prediction for point $z$.

Then RMSE is defined as

$$\text{RMSE} = \sqrt{\frac{1}{N(T - t_0)} \sum_{i,t} (z_{i,t} - \hat{z}_{i,t})^2}$$

ND (Normalized Deviation) is calculated as

$$\text{ND} = \frac{\sum_{i,t} |z_{i,t} - \hat{z}_{i,t}|}{\sum_{i,t} |z_{i,t}|}$$

## F.4  LL, Q-LOSS, AND D-LOSS (IMAGE GENERATION)

Log Likelihood and Q-Loss (quantile loss as defined in our paper) with respect to the digits generation task are optimization objectives and not metrics. We measure these with to depict how one is affected in the same model trained on the other loss function.

D-Loss ($D : \mathbb{R}^n \to \{0, 1\}$) is measured as the accuracy with which a SVM can discern the difference between generated samples S and true samples Y, which are labeled 0 and 1 respectively.

Given validation set with $m$ data points and labels $X \in \{0, 1\}$, we have that

$$\text{D-Loss} = \frac{1}{m} \sum_i^m \mathbf{D}(x'^{(i)}) x^{(i)} + (1 - \mathbf{D}(x'^{(i)}))(1 - x^{(i)})$$

For MNIST, we trained autoregressive models with the PixelCNN architecture, Each model was trained for 300 epochs with a learning rate of 1e-3, and has hidden layer size 784.

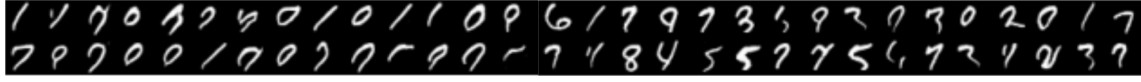

  (a) PixelMAF-LL                                          (b) PixelMAF-QL.

Figure 7: MNIST samples

Table 11: MNIST Generation Experiments

| Method | U-CRPS | CRPS | NLL |
|---|---|---|---|
| PixelMAF-LL | .128 | .279 | -6011 |
| PixelMAF-QL | .099 | .215 | -5888 |

**Results.**   From the samples in Figure 7, the results generated from the MAF-QL seems to be much more visually accurate than MAF-LL, which is trained on a LL-based loss, even though they have the same architecture.

