# OpenReview forum: "Autoregressive Quantile Flows for Predictive Uncertainty Estimation"
_ICLR.cc/2022/Conference — ICLR 2022 Spotlight_

### Official Review · Reviewer_AxZh · 2021-10-27

**Correctness:** 3
**Technical Novelty And Significance:** 2
**Empirical Novelty And Significance:** 3
**Recommendation:** 6
**Confidence:** 4

**Main Review:**

Overall, the proposed autoregressive quantile flow is a reasonable application of normalizing flow in quantile regression. It is also extendable to various different flow transformations and applicable to a wide variety of regression problems. However, this paper has not provided a very clear and strong motivation of why using normalizing flow (or autoregressive flow in particular) for uncertainty estimation, given there are many models that can be trained by gradient-based methods, one can simply replace their objective functions by quantile/CRPS loss. By choosing an appropriate model according to data property, the performance should also be no worse than normalizing flow. Moreover, many other approaches (e.g., diffusion model) also support forward and reverse training, which is not a unique feature of this method. In addition, the authors may also want to explain the procedure more clearly by adding an algorithm box into the main paper.

**Summary Of The Paper:**

This paper proposed a quantile regression method for uncertainty estimation based on autoregressive quantile flow. The flow model can be trained in both forward and reverse setting using different loss functions, and the quantile flow framework can be combined with other linear or non-linear transformations. The authors have conducted diverse empirical evaluations on objective detection (bounding box regression), time series forecasting, and generative models to demonstrate the advantage of the proposed method.

**Summary Of The Review:**

This paper has proposed a normalizing flow-based method to address the predictive uncertainty in regression problems. The significance of novelty can be further improved before it can be accepted.

## Update after Rebuttal

The authors have addressed most of the concerns. It would be better to clarify the difference with diffusion models in the main paper. Good luck!

---

> ### Author Response · Authors · 2021-11-22
> **Feedback to reviewer AxZh**
>
> We thank the reviewer for the constructive feedback. We have listed the responses  to each of the concerns below:
>
> Q1. This paper has not provided a very clear and strong motivation of why using normalizing flow (or autoregressive flow in particular) for uncertainty estimation, given there are many models that can be trained by gradient-based methods, one can simply replace their objective functions by quantile/CRPS loss.
>
> A: We have tested our models on various synthetic datasets, and tables and procedures can be found in the appendix. In summary, we generated some data on multidimensional gaussian experiments, and measured our AQF against a standard gaussian as well as a non-autoregressive-flow network which is trained using quantile loss (Table 10). Though the quantile loss objective replacement does provide improvement over the gaussian assumptions, it still falls short in comparison to the performance of AQF. We have also included metrics of points sampled from the true distribution as a comparative performance to the various methods as an upper bound on performance.
>
> However, experiments aside, the reviewer does seem to have misunderstood the main contributions and objectives of the paper. Rather than considering it in context of a paper in the field of uncertainty estimation, it is better considered as a general contribution to flow architecture in particular, so saying that it has limited novelty is somewhat excessive. In this regard, it would be bold to compare our method against every possible generative model available, in comparison to an evaluation against other flow-architectures as we have done in the paper.
>
> Q2: Moreover, many other approaches (e.g., diffusion model) also support forward and reverse training, which is not a unique feature of this method.
>
> A: The “forward and reverse” denoted in our paper is a different meaning from the “forward and reverse” training denoted in diffusion models. Our definition of reverse was that, by training the model in the reverse direction, we can recover the same setup as what is used in NAF (which seeks to transform the true distribution into a Gaussian), in comparison to the diffusion model in which the “reverse” direction is used to recover the original image from the generated gaussian noise image. The reverse direction in diffusion methods is essential in completing the model, while in our proposed method, the reverse direction can be regarded as a plug-in module.
> In addition, parametrizing the noise in the quantile format allows us to tractably control the noise, as compared to diffusion models which can only sample from gaussian noise inputs. In addition, the flexibility of quantile flows allows it to adapt to further developments in neural architectures and frameworks in the flow area.
>
> Q3: In addition, the authors may also want to explain the procedure more clearly by adding an algorithm box into the main paper.
>
> A: We have included an algorithm box in the appendix.

---

> ### Author Response · Authors · 2021-11-29
> **Thanks for the update!**
>
> Thanks so much for the update! We'll make sure to clarify the difference with diffusion models in the main paper.

---

### Official Review · Reviewer_85oZ · 2021-10-31

**Correctness:** 3
**Technical Novelty And Significance:** 4
**Empirical Novelty And Significance:** 3
**Recommendation:** 8
**Confidence:** 4

**Main Review:**

The strengths of this article are as follows:
1. The proposed objective is novel, which does not need to compute the inverse of the Jacobian, which is intractable in most cases. Moreover, it enables complicated transformers like deep neural networks to be applied in flow models and simplifies the implementation, thus having good potential for future applications.
2. Benefit from the design of objective functions, the proposed framework supports a sampling-based training paradigm. Such a mechanism embraces a broader range of loss functions to the flow framework.
3. It is also interesting to see that the proposed quantile flows could also provide the distribution estimation of model outputs, which can be applied to provide uncertainty of predictions.

Despite the main contributions of the article, it could still be improved from the following aspects:
1. Providing more detailed descriptions of the evaluation metrics for better understandability.
2. Some typos are found:
  - In the second paragraph of subsection 2.2, “the quality of forecasts if often...” -> “the quality of forecasts is often...”.
  - In eq. (1), “F_\theta (y_i)” -> “F_\theta (y)”.
  - In the third paragraph of subsection 6.1, a period is missing at the end of the first sentence.
3. Please cite the source of datasets or provide links. Also, please unify the reference format and cite the following paper:
Sandler, Mark, Andrew Howard, Menglong Zhu, Andrey Zhmoginov, and Liang-Chieh Chen. "Mobilenetv2: Inverted residuals and linear bottlenecks." The IEEE conference on computer vision and pattern recognition, pp. 4510-4520. 2018.


**Summary Of The Paper:**

This paper proposes a novel framework for training flow models named Autoregressive Quantile Flows (AQF). The proposed method utilizes a new objective by evaluating forecasts with proper scoring rules, including the continuous ranked probability score and the check score. The advantages of the proposed objective are 1) it could avoid the explicit calculation of the determinant of the Jacobian matrix and 2) it could also provide uncertainty estimation for predictions. Experiments on multiple tasks including regression, object detection, time series forecasting, and generation validate the effectiveness of this framework.

**Summary Of The Review:**

This article provides a novel and interesting idea to improve flow models in multiple aspects. I think this work can inspire other researchers in the community, so I tend to accept the paper after some minor modifications.

Post-Rebuttal
----
The paper has its merits and the authors have revised the manuscript according to the comments.

Thus, I keep the score and recommand acceptance.

---

> ### Author Response · Authors · 2021-11-22
> **Feedback to reviewer 85oZ**
>
> We thank the reviewer for the constructive feedback. Below we have listed our responses to each of the comments:
>
> Q1: Providing more detailed descriptions of the evaluation metrics for better understandability.
>
> A: Thanks so much for your input! We have revised our paper to explain those more clearly, along with a metric definitions section in the appendix to go more into depth with them.
>
> Q2: Some typos are found:
>
> A: Thank you for the notes! We have proofread the paper more to iron out the typos.
>
> Q3: Please cite the source of datasets or provide links. Also, please unify the reference format and cite the following paper: Sandler, Mark, Andrew Howard, Menglong Zhu, Andrey Zhmoginov, and Liang-Chieh Chen. "Mobilenetv2: Inverted residuals and linear bottlenecks." The IEEE conference on computer vision and pattern recognition, pp. 4510-4520. 2018.
>
> A; Thank you for the suggestion! We have included the citation on our paper and fixed our other citations.

---

> ### Author Response · Authors · 2021-11-29
> **Thanks for the update!**
>
> Thanks so much for the post-rebuttal response!

---

### Official Review · Reviewer_UtzJ · 2021-11-01

**Correctness:** 3
**Technical Novelty And Significance:** 4
**Empirical Novelty And Significance:** 3
**Recommendation:** 8
**Confidence:** 4

**Main Review:**

Strong points:
- The proposed objective functions (quantile loss and continuous ranked probability score) appear well-motivated and have nice theoretical properties.
- The generalization to quantile flows is well-motivated and well-described.
- The implementation of all methods is clearly described.
- The paper is well-structured and introduces complex topics well.
- The analyses of the UCI datasets are comprehensive.

Major concerns:
- Since the manuscript proposes several new objective functions and architectures, it would be nice to have comprehensive experiments on synthetic data. In particular,
    - how well does the method for quantile function regression capture the true quantile function in a univariate case?
    - do the prediction intervals behave as expected from the theory?
    - how well does the method estimate the cdf at various points (e.g., the median, the 25th and 75th percentiles)?
    - compare this method vs maximum likelihood in terms of timing and accuracy
- Some of the terms aren't defined (e.g., the column headings in Table 3, Table 4)

Minor concerns:
- check throughout for typos

**Summary Of The Paper:**

The paper proposes an interesting approach for estimating densities and providing uncertainty estimates for these densities. The paper extends normalizing flows in several ways: (1) by using objective functions based on proper scoring; (2) by using autoregressive quantile flows; and (3) by defining quantile flow regression.

**Summary Of The Review:**

I vote for accepting the paper subject to some additional experiments.

---

### Public Comment · ~Kashif_Rasul1 · 2021-11-15
**Proposition 1**

The source for the proof of prop 1 can be found in the references of our IQN-RNN paper, namely: Laio, F. and Tamea, S. Verification tools for probabilistic forecasts of continuous hydrological variables. from 2007.

```
@Article{hess-11-1267-2007,
AUTHOR = {Laio, F. and Tamea, S.},
TITLE = {Verification tools for probabilistic forecasts of continuous hydrological variables},
JOURNAL = {Hydrology and Earth System Sciences},
VOLUME = {11},
YEAR = {2007},
NUMBER = {4},
PAGES = {1267--1277},
URL = {https://hess.copernicus.org/articles/11/1267/2007/},
DOI = {10.5194/hess-11-1267-2007}
}
```

---

> ### Author Response · Authors · 2021-11-22
> **Proposition 1**
>
> Thanks so much for bringing this to our attention! We have additionally cited the work in the paper.

---

### Author Response · Authors · 2021-11-23
**Rebuttal Draft Revision**

Thanks so much to all the reviewers for your thoughtful feedback!

According to the comments, we have made the following changes to the paper to address the concerns:

1. Added extensive synthetic experimentation in the appendix covering 1-dimensional as well as multidimensional synthetic experiments. These include an analysis of confidence interval quality at key intervals, as well as comparison of quantile flow architecture against non-autoregressive-flow architecture using quantile loss.
2. Added formal definitions to the appendix for the metrics we use in the paper which are not closely addressed in the main body of the paper itself
3. Added an algorithm box for the AQF sampling procedure
4. Fixed citations and typos which are in the paper

---

### Decision · Program_Chairs · 2022-01-20

**Decision:**

Accept (Spotlight)

**Comment:**

The paper proposes a framework for training autoregressive flows based on proper scoring rules. The proposed framework is shown to be a computationally appealing alternative to maximum-likelihood training, and is empirically validated in a wide variety of applications.

All three reviewers are positive about the paper and recommend acceptance (one weak, two strong). The reviewers describe the paper as well written and well motivated, and recognize the paper's contribution as significant.

Overall, this is a nice and promising methodological exploration of flow-model training that is worth communicating to the ICLR community.